# Association between Early Immune-Related Adverse Events and Survival in Patients Treated with PD-1/PD-L1 Inhibitors

**DOI:** 10.3390/jcm12030736

**Published:** 2023-01-17

**Authors:** You-Cheng Zhang, Tian-Chen Zhu, Run-Cong Nie, Liang-He Lu, Zhi-Cheng Xiang, Dan Xie, Rong-Zhen Luo, Mu-Yan Cai

**Affiliations:** 1State Key Laboratory of Oncology in South China, Collaborative Innovation Center for Cancer Medicine, Sun Yat-sen University Cancer Center, Guangzhou 510060, China; 2Department of Pathology, Sun Yat-sen University Cancer Center, Guangzhou 510060, China

**Keywords:** immune-related adverse events, efficacy, immune checkpoint inhibitors, tumor, meta-analysis

## Abstract

Background: Immune-related adverse events (irAEs) are side effects that reflect the activation of patients’ immune systems after treatment with immune checkpoint inhibitors (ICIs). However, there is no meta-analysis on the effect of early irAEs on patient survival. Thus, we assessed the association between early irAEs and the survival of patients treated with ICIs. Methods: PubMed, Embase, and Web of Science were searched from May 2010 to May 2020 for all the retrospective and prospective comparative studies to evaluate the hazard ratios (HRs) for death. A random-effects model was used to calculate the pooled HR for death, and heterogeneity was assessed using I² statistics. The main outcomes were overall survival (OS) and progression-free survival (PFS). Results: A total of 11 reports with 2077 patients were included. A significant association was observed between early irAEs and a favorable clinical outcome. Patients with early irAEs had prolonged OS (HR: 0.62, 95% confidence interval (CI): 0.53–0.74, *p* < 0.001) and PFS (HR: 0.53, 95% CI: 0.41–0.66, *p* < 0.001) compared to those without; these results were confirmed using a sensitivity analysis. The irAE types, malignancy types, and sample size were correlated with patients’ clinical outcomes. Conclusions: Early irAEs, especially cutaneous irAEs, correlated with a better clinical outcome in patients treated with ICIs.

## 1. Background

Immune checkpoint inhibitors (ICIs) are promising therapeutic options for various malignancies [1]. ICIs restore the ability of T cells to selectively recognize and eliminate cancer cells by blocking immune checkpoint signals. However, there is great heterogeneity in treatment efficacy among patients who received ICIs, and effective biomarkers are needed for predicting patients’ responses to ICIs. Immune-related adverse events (irAEs) are side effects that reflect the activation of patients’ immune systems after treatment with ICIs. Thus, they are a potential predictor for the response to immunotherapy. Various irAEs have been reported in patients receiving ICIs, such as skin reactions, thyroid dysfunction, pneumonitis, and hepatitis [2,3,4,5,6,7]. The severity of irAEs ranges from mild and manageable to severe and life-threatening if they are not diagnosed early and treated appropriately [8]. The underlying mechanisms of irAEs include the bystander effect, shared epitopes, and gut microbiome composition [9].

Existing systematic reviews and meta-analyses have confirmed that irAEs are correlated with favorable clinical outcomes and may be useful biomarkers in clinical practice. Subgroup analyses were conducted to investigate the main characteristics of irAEs that may affect ICI efficacy, such as irAE site and severity [10,11,12]. However, there are few studies on the time of irAE onset, which may also be a main characteristic of irAEs. We hypothesized that the time of irAE onset has a potential connection to the efficacy of immunotherapy. Patients with early disease progression in whom anti-programmed cell death (PD)-1 treatment is interrupted are exposed to the potential “triggering effect” for a shorter time than those without; therefore, they have a lower risk for irAEs [13].

Several clinical studies have reported an association between early irAEs and ICI efficacy [14,15]. The definition of early irAEs is controversial [2,13,16,17,18,19,20,21,22,23]. Though prior studies have shown a good efficacy of ICIs in patients who developed irAEs, there is no meta-analysis on the effect of early irAEs on patient survival. Furthermore, whether the development of dermatological irAEs after treatment with ICIs can predict a favorable clinical outcome is unclear. Thus, we systemically reviewed and analyzed the available literature to pool the results and explore the effect of time of onset of irAEs and patient survival.

## 2. Materials and Methods

This study was performed according to the Preferred Reporting Items for Systematic Reviews and Meta-analyses statement [24].

### 2.1. Literature Search Strategy

Two authors conducted a comprehensive systematic search of PubMed, Embase, and Web of Science with no language restrictions from 2010 to May 2020 for clinical studies of anti-PD-1/ PD ligand 1 (PD-L1) that reported the association between the time of onset of irAEs and patient outcomes. The detailed search procedure is described in Appendix A.

### 2.2. Inclusion and Exclusion Criteria

The inclusion criteria were as follows: 1. studies involving patients with advanced solid tumors irrespective of tumor site and who received PD-1/PD-L1; 2. studies on the association between irAE occurrence and ICI efficacy in patients with cancer including hazard ratios (HRs) of the overall survival (OS) and progression-free survival (PFS) of patients who developed irAEs and those who did not; and 3. studies with a median onset time of irAEs within two months after immunotherapy initiation or studies whose landmark analysis was conducted within two months.

Studies involving combination therapy and anti-CTLA-4 monotherapy were excluded because of a greater heterogeneity in the irAEs in these patients than in those who received anti-PD-1/PD-L1 monotherapy [11]. Editorials, letters to the editor, review articles, case reports, and animal experimental studies were excluded. When different reports published the same population data, the most recent or complete report was included in our study.

After the first selection of studies, all the references were screened from the included articles for any further eligible publications. According to previously published articles, irAEs that develop within two months after the commencement of ICIs were defined as early irAEs.

### 2.3. Data Extraction

Data from the included studies were extracted and summarized independently by two authors (YCZ and LHL). Any disagreement was resolved by the senior author (RCN). The primary outcome was OS, and the secondary outcome was PFS. The reported HRs for OS, PF, and the following clinicopathological characteristics of each eligible trial were extracted: title, author, publication year, cancer type, agent, landmark analysis, irAE type, grade of irAE, and trial design.

If a study had only survival curves but no HR data, the Engauge Digitizer and Hazard Ratio Meta-analysis Spreadsheet, developed by Hans Messersmith, was utilized to extract HR data from the survival curves [25].

### 2.4. Quality Assessment and Statistical Analyses

The methodological quality of the studies was assessed using the Newcastle–Ottawa scale, which consists of three factors: patient selection, comparability of the study groups, and assessment of outcomes. A score of 0–9 (allocated as stars) was assigned to each study. All meta-analyses were performed using STATASE 12.0. The HR was used to compare variables. All results were presented with 95% confidence intervals (CIs).

Statistical heterogeneity between different studies was assessed using the chi-square test, with the level of significance set at *p* < 0.10. Heterogeneity was quantified using the I² statistic. Heterogeneity was considered substantial when I² was >50%. A random-effects model was used for data analysis due to the moderate heterogeneity between different studies.

Subgroup analyses were performed to investigate the effect of early dermatological irAEs on anti-PD-1/PD-L1 efficacy. Potential publication bias was assessed using visual inspection of a funnel plot and was evaluated using Begg’s regression asymmetry tests. A *p*-value <0.05 was considered statistically significant.

## 3. Results

### 3.1. Literature Search

We identified 745 relevant studies. A total of 11 studies with 2077 patients satisfied the inclusion criteria and were included in the final analysis. The detailed study selection process is summarized in Figure 1.

### 3.2. Study Characteristics

The detailed characteristics of the included studies are presented in Table 1. Among these included studies, there were nine retrospective studies and two prospective studies. Specifically, Hosoya et al. conducted trials in retrospective and prospective cohorts. The included malignancies were non-small cell lung cancer (NSCLC, n = 8), renal cell carcinoma (n = 2), and advanced gastric cancer (n = 1). The sample sizes ranged from 43 to 559. The median follow-up interval varied from 9.9 to 32 months. The methodology quality of the included trials was generally moderate to good, and the main issue that affected the quality was the follow-up duration (Appendix A).

### 3.3. Correlation of Early irAEs and Treatment Efficacy

OS (HR: 0.48, 95% CI: 0.39–0.56, *p* < 0.001) and PFS (HR: 0.53, 95% CI: 0.41–0.66, *p* < 0.001) (Figure 2) were higher in the patients who developed early irAEs than in those who did not. Moderate heterogeneity was observed in the overall treatment effect across the 11 comparisons (*p* = 0.054, *I*² = 42.0%); therefore, the random-effects model was preferred for the pooled analysis.

### 3.4. Subgroup and Sensitivity Analyses

A subgroup analysis was performed based on the irAE type, tumor type, and sample size. Early dermatological irAEs were associated with significantly prolonged OS (HR: 0.51, 95% CI: 0.27–0.75, *p* < 0.001) and PFS (HR: 0.58, 95% CI: 0.41–0.75; *p* < 0.001; Figure 3). The sample sizes were not significantly correlated with patient prognosis (Figure 4). The stratified analysis of tumor type showed a favorable clinical outcome for OS (HR: 0.47, 95% CI: 0.38–0.57, *p* < 0.001) and PFS (HR: 0.55, 95% CI: 0.41–0.69, *p* < 0.001) in patients with NSCLC (Figure 5). The pooled results for both OS and PFS remained significant in the sensitivity analysis, regardless of the study, indicating a robust association between the development of early irAEs and prolonged OS and PFS (Appendix A).

### 3.5. Publication Bias

With an even distribution around the vertical axis, there was no substantial asymmetry in the visual inspection of Begg’s funnel plot (*p* = 0.653, Appendix A). This was confirmed with Egger’s tests (*p* = 0.412), indicating no obvious publication bias.

## 4. Discussion

PD-1/PD-L1 inhibitors have revolutionized the treatment of various human cancers. Studies on irAEs developed during treatment with PD-1/PD-L1 inhibitors are critical for clinical decision-making. In our meta-analysis, the correlation between the efficacy of PD-1/PD-L1 inhibitors and early-onset irAEs was analyzed. We concluded that the occurrence of early irAEs is linked to better clinical outcomes, particularly cutaneous irAEs. Further subgroup analysis of tumor types and sample sizes demonstrated similar findings.

Previous studies have shown a positive association between irAE development and a favorable survival benefit. Specifically, Zhou et al. [10] performed a meta-analysis to pool the predictive effects of different irAE types and grades. The authors reported that the occurrence of endocrine, dermatological, and low-grade irAEs was significantly associated with a better clinical outcome in patients treated with ICIs. The time of onset of irAE is also a critical factor for the correlation between irAEs and clinical outcomes of treatment with ICI. Patients who develop irAEs are those who receive treatment with ICI for longer periods, and, thus, have a better prognosis than those who do not [13]. Thus, the so-called ‘guarantee-time bias’ may be induced if we do not consider the time of irAEs [26]. However, the reported time of onset of irAEs varies.

In our study, the development of early irAEs correlated significantly with prolonged PFS and OS. This makes sense because: (1) the occurrence of early irAEs reflects the function of T cells. According to the shared epitope theory, blockade of the PD-1/PD-L1 axis activates T cells wherein auto-antigens from normal tissues are eliminated at the same time. Thus, it is reasonable to consider that tumor cells are eliminated by activated T cells. (2) Patients who developed disease early after PD-1/PD-L1 inhibitor initiation often have a poor physical state, resulting in the discontinuation of anti-PD-1/PD-L1 therapy. Thus, these patients have little chance of developing irAEs because they only receive therapy for a short time. We also found that dermatological irAEs are associated with favorable outcomes. This can be attributed to the different severities of irAEs in different organs. Generally, dermatological irAEs are mild and manageable. In contrast, other types of irAEs, such as pulmonary and hepatobiliary irAEs, can be severe and even fatal. Patients developing these irAEs often have a poor physical state and are thus more likely to experience severe disease or death.

The immune checkpoint blockade induces the emergence of irAEs by unbalancing the immune system, although the exact mechanisms of the development of irAEs are still unclear. Cross-reactivity of the neoantigen-directed antitumor response with normal proteins may be one of the underlying mechanisms [27]. During checkpoint inhibitor therapy, T cells recognize the shared antigens and simultaneously target tumor cells and normal tissues [28]. CD8+ cytotoxic T lymphocyte-mediated cell lysis induces the release of neoantigens, tumor antigens, and auto-antigens, which are processed and presented by antigen-presenting cells from normal tissues [29]. This phenomenon of “epitope spreading” leads to the diversification of the T cell repertoire and reduces immune tolerance [30]. Thus, tumor regression is associated with the development of irAEs, and ICIs enhance the antitumor immune response as indicated by the presence of autoantibodies. The occurrence of irAEs is also thought to represent the bystander effects from activated T cells [9,27]. Normal tissue may experience severe inflammation when key negative regulators of T cell function are removed using ICI therapy. However, this is controversial because the previous meta-analysis reported a better ICI efficacy in patients with low-grade irAEs [10]. The composition of the gut microbiome may also play an important role in the efficacy of ICI therapy. Patients with specific microbiomes had improved clinical outcomes and ICI-induced colitis [31].

There are some limitations to our study. First, this is a meta-analysis, and patient-related variables were not included in the analysis. Second, the inherent limitations of the included studies limit the generalizability of our results. Third, some of the eligible trials had limited participants with a short follow-up duration, which may have resulted in the wide CIs of the HRs of the treatment effects, thereby confounding our pooled HRs. Finally, most of the studies were retrospective, which may increase the risk of bias.

In conclusion, early irAEs correlated significantly with prolonged OS and PFS in patients who received anti-PD-1/PD-L1 therapy. Future well-designed prospective studies are warranted to confirm our findings.

## Figures and Tables

**Figure 1 jcm-12-00736-f001:**
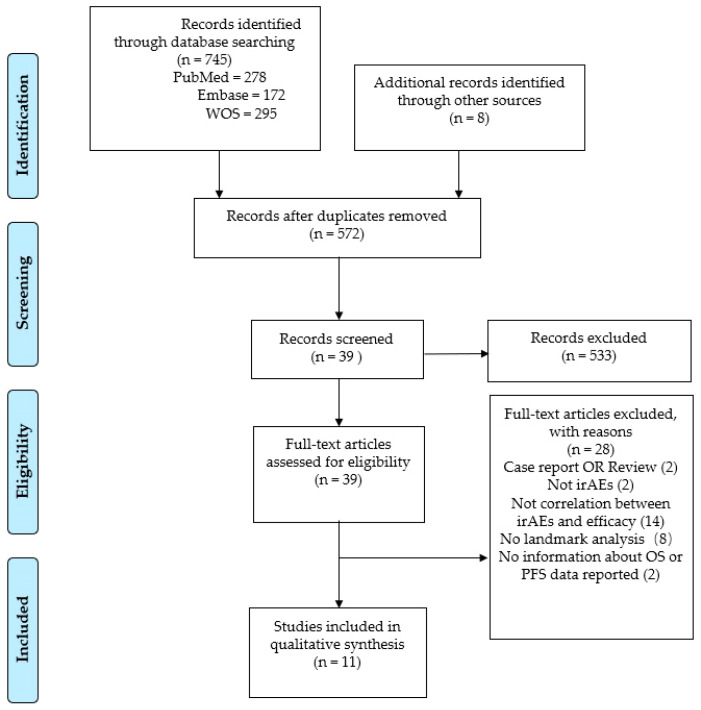
The flow diagram.

**Figure 2 jcm-12-00736-f002:**
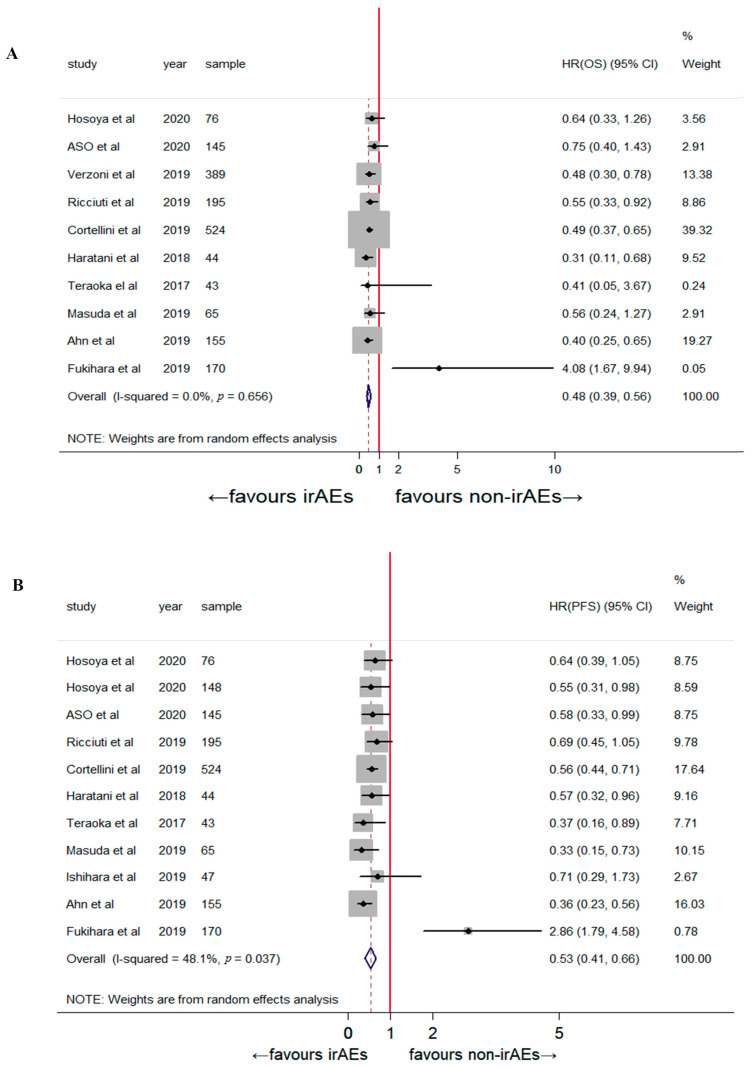
The association between early-onset irAEs and (**A**) overall survival [2,8,13,14,15,16,17,20,21,22] and (**B**) progression-free survival [2,8,13,14,15,16,17,18,20,21].

**Figure 3 jcm-12-00736-f003:**
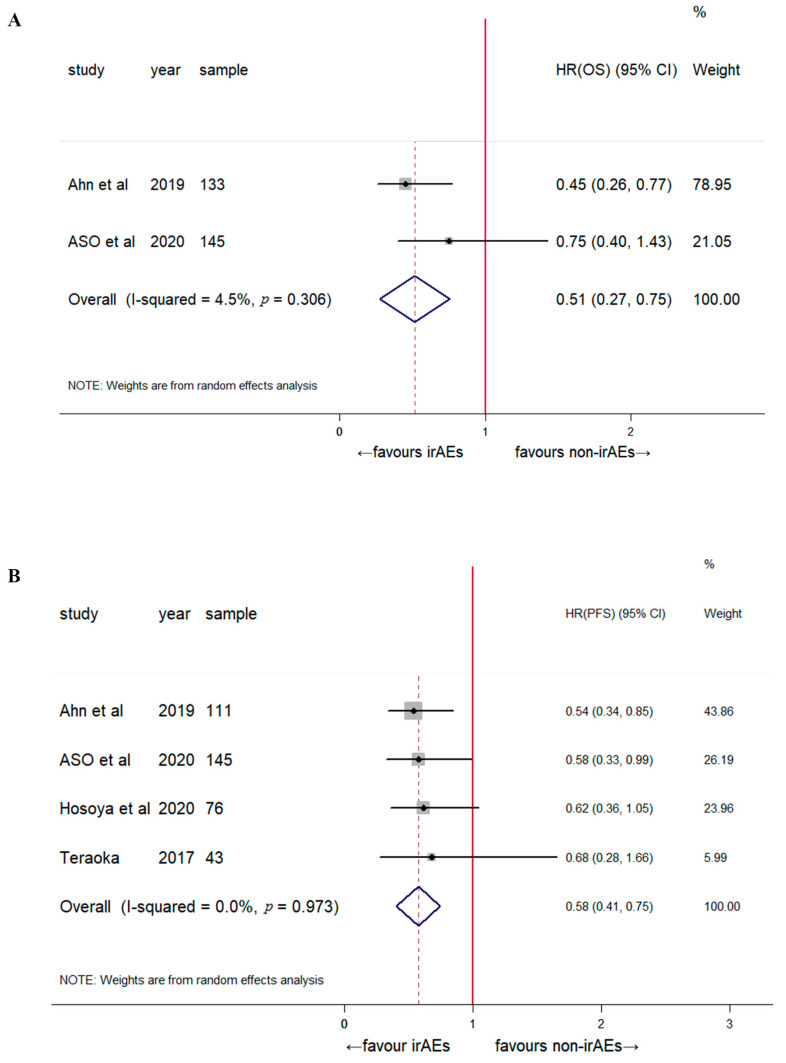
Subgroup analysis of the association between early-onset dermatological irAEs and the clinical outcome of patients treated with immune checkpoint blockade. Overall survival (**A**) [2,16] and progression-free survival (**B**) [2,14,15,16].

**Figure 4 jcm-12-00736-f004:**
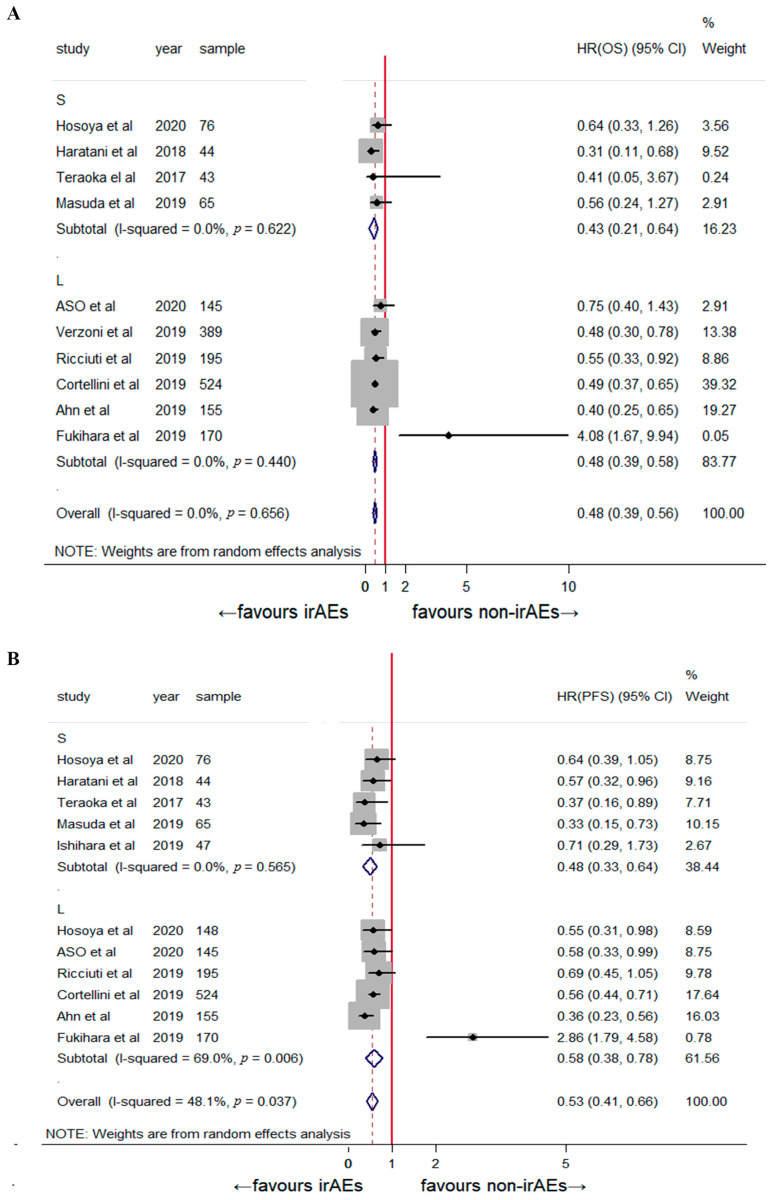
Subgroup analysis of the association between early-onset irAEs and (**A**) overall survival [2,8,13,14,15,16,17,20,21,22] and (**B**) progression-free survival [2,8,13,14,15,16,17,18,20,21] stratified by sample size. S indicates small; L indicates large.

**Figure 5 jcm-12-00736-f005:**
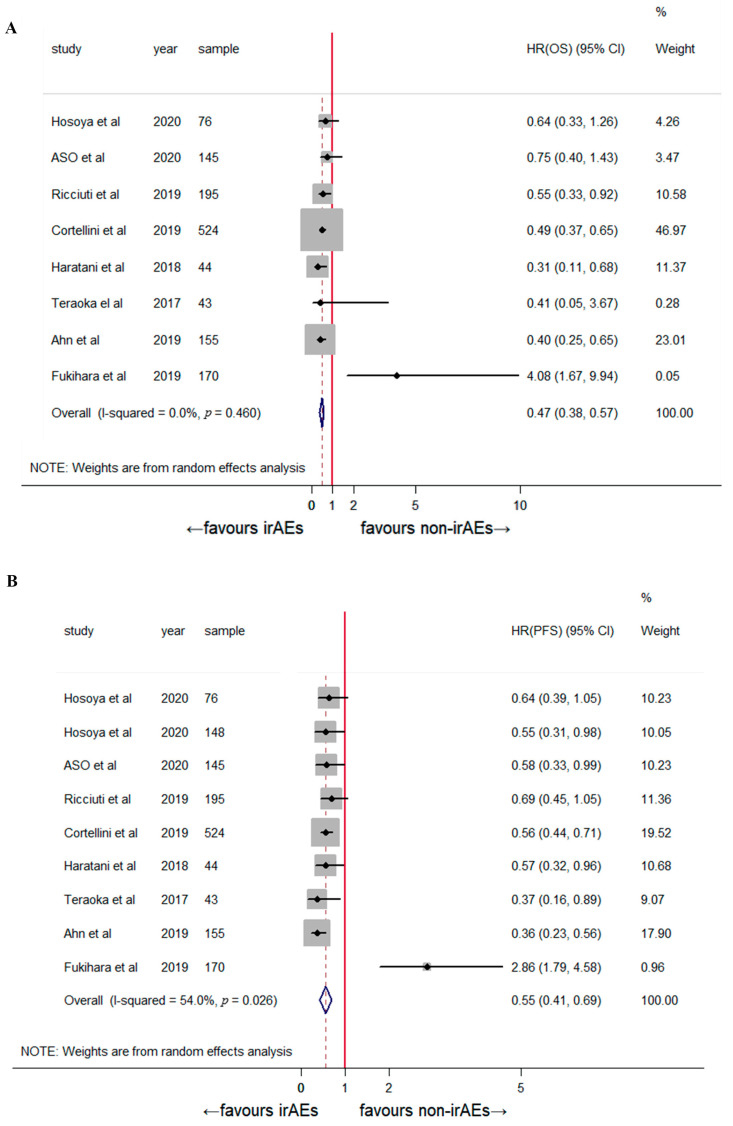
Subgroup analysis of the association between early-onset irAEs and overall survival (**A**) [2,8,13,14,15,16,17,21] and progression-free survival (**B**) [2,8,13,14,15,16,17,21] in patients with non-small cell lung cancer.

**Table 1 jcm-12-00736-t001:** Main characteristics of the eligible studies.

Study	Cancer Type	Agents	Exposed Group/Total, No.	irAE Type	irAE Grade	Hazard Ratio (95% CI)	Landmark Analysis	Design
Aso, 2020 [2]	NSCLC	N/P	51/155	skin	1–5	PFS:0.58 (0.33–0.99)	6 weeks	RC
						OS:0.75 (0.40–1.43)		
Haratani, 2018 [8]	NSCLC	N	69/134	Global	1–4	PFS:0.57 (0.33–0.96)	6 weeks	RC
						OS:0.31 (0.11–0.68)		
Hosoya, 2020 [15]	NSCLC	N/P	37/76	Global	1–4	PFS:0.64 (0.39–1.05)	6 weeks	PC
						OS:0.64 (0.33–1.26)		
						PFS:0.55 (0.31–0.98)	3 weeks	RC
Ricciuti, 2019 [21]	NSCLC	N	85/195	Global	1–4	PFS:0.69 (0.45–1.05)	6 weeks	RC
						OS:0.55 (0.33–0.92)		
Cortellini, 2019 [13]	NSCLC	N/P	231/559	Global	1–4	PFS:0.56 (0.44–0.71)	6 weeks	RC
						OS:0.49 (0.37–0.65)		
Ahn, 2019 [16]	NSCLC	N/P	55/155	Global	1–4	PFS:0.36 (0.23–0.56)	6 weeks	RC
						OS:0.40 (0.25–0.65)		
Verzoni, 2019 [22]	RCC	N	76/389	Global	1–4	OS:0.48 (0.30–0.78)	6 weeks	
Ishihara, 2019 [18]	RCC	N	23/47	Global	1–4	PFS:0.71 (0.29–1.73)	2-cycle	RC
Masuda, 2019 [20]	AGC	N	14/65	Global	1–4	PFS:0.33 (0.15–0.73)	2 months	RC
						OS:0.56 (0.24–1.27)		
Teraoka, 2017 [14]	NSCLC	N	19/43	Global	1–2	PFS:0.37 (0.16–0.89)	6 weeks	PC
						OS:0.41 (0.05–3.67)		
Fukihara, 2019 [17]	NSCLC	N/P	27/170	pulmonary	1–5	PFS:2.86 (1.79–4.58)	6 weeks	RC
						OS:4.08 (1.67–9.94)		

Abbreviations: irAE: immune-related adverse event, NSCLC: non-small-cell lung carcinoma, RCC: renal cell carcinoma, AGC: advanced gastric cancer, RC: retrospective cohort, PC: prospective cohort, N: nivolumab, P: pembrolizumab, OS: overall survival, PFS: progression-free survival.

## Data Availability

This meta-analysis was not registered. A protocol was not prepared. No new data were created or analyzed in this study. Data sharing is not applicable to this article.

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
