# Peer review of "Association between Early Immune-Related Adverse Events and Survival in Patients Treated with PD-1/PD-L1 Inhibitors"

_jcm, 2023, doi:10.3390/jcm12030736_

Round 1

Reviewer 1 Report

This metaanalysis is clearly reported. I would like a comment on the study of Fukihara, 2019 whose results are really different form those obtained in the other studies, probably because of the particular irAE they consider. 

I wonder how authors could estimate Kaplan-Meier curves if they performed a meta analysis based on published data. I guess they only have Hazard Ratio, though interesting, no Kaplan-Meier is possible without individual patients data.

Reviewer 2 Report

The manuscript entitled “Association between early immune-related adverse events and survival in patients treated with PD-1/PD-L1 inhibitors” by You-Cheng Zhang is based on a meta-analysis carried out in cancer patients treated with PD-1/PD-L1 inhibitors. Authors show significant association between early Immune-related adverse events (irAEs) adverse events and overall survival of patients. This is based on the Hazard ratio calculated. Additionally. it would be interesting to know the survival based on standard Kaplan Meir analysis by dichotomizing at the median fold changes of (irAEs) (post/pre) and clinical outcome of the patients.
